# Proximity Mapping of CCP6 Reveals Its Association with Centrosome Organization and Cilium Assembly

**DOI:** 10.3390/ijms24021273

**Published:** 2023-01-09

**Authors:** Sergi Rodriguez-Calado, Petra Van Damme, Francesc Xavier Avilés, Ana Paula Candiota, Sebastian Tanco, Julia Lorenzo

**Affiliations:** 1Institut de Biotecnologia i Biomedicina, Departament de Bioquímica i Biologia Molecular, Universitat Autònoma de Barcelona, 08193 Cerdanyola del Vallès, Barcelona, Spain; 2iRIP Unit, Laboratory of Microbiology, Department of Biochemistry and Microbiology, Ghent University, K. L. Ledeganckstraat 35, 9000 Ghent, Belgium; 3Centro de Investigación Biomédica en Red en Bioingeniería, Biomateriales y Nanomedicina (CIBER-BBN), 08193 Cerdanyola del Vallès, Barcelona, Spain

**Keywords:** cytosolic carboxypeptidases, CCP6, polyglutamylation, proteomics, biotin-dependent proximity labeling (BioID), BirA*, centrioles, cilia, Joubert syndrome, cilium assembly

## Abstract

The cytosolic carboxypeptidase 6 (CCP6) catalyzes the deglutamylation of polyglutamate side chains, a post-translational modification that affects proteins such as tubulins or nucleosome assembly proteins. CCP6 is involved in several cell processes, such as spermatogenesis, antiviral activity, embryonic development, and pathologies like renal adenocarcinoma. In the present work, the cellular role of CCP6 has been assessed by BioID, a proximity labeling approach for mapping physiologically relevant protein–protein interactions (PPIs) and bait proximal proteins by mass spectrometry. We used HEK 293 cells stably expressing CCP6-BirA* to identify 37 putative interactors of this enzyme. This list of CCP6 proximal proteins displayed enrichment of proteins associated with the centrosome and centriolar satellites, indicating that CCP6 could be present in the pericentriolar material. In addition, we identified cilium assembly-related proteins as putative interactors of CCP6. In addition, the CCP6 proximal partner list included five proteins associated with the Joubert syndrome, a ciliopathy linked to defects in polyglutamylation. Using the proximity ligation assay (PLA), we show that PCM1, PIBF1, and NudC are true CCP6 physical interactors. Therefore, the BioID methodology confirms the location and possible functional role of CCP6 in centrosomes and centrioles, as well as in the formation and maintenance of primary cilia.

## 1. Introduction

Polyglutamylation is a reversible posttranslational protein modification (PTM) of the γ-carboxyl groups of glutamate in a primary protein sequence by peptide-like bond linkage with the amino group of free glutamate, and further extendible to polyglutamate chains of variable length in target proteins [1,2,3]. Tubulins and nucleosome assembly proteins (NAPs) are well-known polyglutamylation targets [4,5]; however, recent studies have expanded the number of potentially polyglutamylated candidates to include proteins such as nucleoplasmin (NPM) [6], the DnaJ homolog subfamily C member 7 (DNJC7) [7] and the mitotic arrest deficient 2 (MAD2) [8]. The length of the glutamate side chains ranges from 1 to 20 residues, enabling the coding of diverse signals and thus the regulation of protein function [9]. For instance, tubulin polyglutamylation regulates the interaction between microtubules and microtubule-associated proteins (MAPs), modulating microtubule-related processes such as centrosome stability, cilia and flagella motility, and neurite outgrowth [2,3,10,11,12,13,14]. Besides its role in tubulin regulation, polyglutamylation has been also associated with other cellular processes such as chromatin remodeling or hematopoiesis [8,15]. Alterations in protein polyglutamylation levels, thus, have been linked to several pathologies including neurodegenerative processes (e.g., cerebellar ataxia caused by increased polyglutamylation levels) [16,17,18] or cancer progression (e.g., cancer cell growth in hepatocellular carcinoma was shown to be promoted by decreased polyglutamylation) [19,20]. Overall, these observations highlight the important role of polyglutamylation in health and disease.

Two enzyme families are involved in the regulation of the length of glutamate side chains. Polyglutamylases, members of the tubulin tyrosine ligase-like (TTLL) enzyme family, catalyze the addition and elongation of polyglutamate chains, whereas deglutamylases catalyze the opposite reaction [21,22,23]. Deglutamylases belong to the family of cytosolic carboxypeptidases (CCPs) and constitute the M14D enzyme subfamily in the MEROPS database of proteases (https://www.ebi.ac.uk/merops/, accessed on 12 December 2022) [24]. The CCP subfamily is composed of six members in humans (i.e., CCP1 to CCP6), all sharing a catalytic carboxypeptidase (CP) domain and a conserved adjacent N-terminal domain of about 150 residues, which is unique to this enzyme family [21,25]. Among them, CCP6 is the shortest CCP member in mammals, formed only by the conserved N-terminal domain and the CP domain. CCP6 is mainly expressed in the central nervous system, testes and kidney, and its presence is also observed in the eye and trachea [22]. At the subcellular level, CCP6 co-localizes with γ-tubulin at centrioles and with the Golgi apparatus protein golgin GM130 during interphase. In addition, CCP6 has also been detected in the basal bodies of ciliated NIH-3T3 cells [21,22,26]. CCP6 functioning as a deglutamylase enzyme has been linked to microtubule regulation, participating in the general assembly of centrioles and basal bodies as well as in flagellar motility [26]. In this context, it is worth mentioning that primary cilia and centrosomes are distinct organelles, which employ the same evolutionary conserved microtubule-based templates (i.e., the centrioles) for their assembly [27]. Cilia are present in nearly all mammalian cells and fulfill a multitude of different functions, as cellular antennae probing the surrounding of the cell for biologically active chemicals, or as motile cilium important for propelling cells (e.g., spermatozoids), or to move the surrounding fluids [28]. As a consequence of the many functions that cilia perform in the body, ciliary dysfunction (by defects in cilium assembly or maintenance) results in diseases that are clinically very diverse and are collectively called ciliopathies [28]. However, the role of CCP6 is not limited to microtubule functioning. For instance, CCP6 has been implicated in the regulation of the innate response against viral DNA by controlling glutamylation levels of cGAS [29], besides playing a critical role in cell reprogramming and early embryonic development by Klf4 deglutamylation in mice [30]. Furthermore, mice lacking CCP6 display a defective megakaryopoiesis process due to the excessive polyglutamylation of MAD2 [8]. In humans, CCP6 has been linked to cancer progression; patients with renal adenocarcinoma frequently display decreased CCP6 levels, leading to the accumulation of the polyglutamylated DNAJC7 and dysregulation of HSP70 and HSP90 proteins [7].

Here, we studied the interactome of CCP6 by means of the recently developed proximity-dependent biotin identification (BioID) approach. The BioID methodology is a unique technique used for screening physiologically relevant protein–protein interactions that occur in living cells [31]. In BioID, the protein of interest is fused to a promiscuous biotin ligase (mutant BirA(R118G), termed BirA*), which upon expression, can biotinylate free lysine residues in nearby or proximal polypeptides that can subsequently be purified and analyzed by mass spectrometry (MS). Since biotin labeling occurs in situ, BioID is particularly suited for interrogating transient interactions, such as the ones occurring between enzymes like CCP6 and its substrates [32,33,34]. We applied BioID in HEK 293 stable cell pools expressing CCP6-BirA*, leading to the identification of 37 putative interactors of CCP6. Our results confirm the association of CCP6 with centrosomes and centrioles and support the role of CCPs in primary cilia formation. Further, the identified CCP6 proximal partners suggest a link between CCP6 and the Joubert syndrome (JS), a ciliopathy that has been related to defects in tubulin polyglutamylation [35,36,37]. All in all, our study provides a set of potential new interactors and substrates for CCP6 that will be valuable toward understanding CCP6 functioning in health and disease.

## 2. Results

### 2.1. Identification of CCP6 Proximal Interactors

In order to achieve a better understanding of the cellular role of CCP6, we used BioID proximity-labeling (Figure 1A) to identify CCP6’s cellular interactome and proximal environment [31]. Therefore, we generated HEK 293 polyclonal stable cell pools expressing CCP6 N-terminally fused to FLAG-tagged BirA*. As control setups, we also applied the BioID workflow on parental Flp-In T-REX 293 cells (i.e., WT cells) as well as on cell pools stably expressing the BirA*-FLAG-tag fused to the C-terminus of the green fluorescent protein (i.e., GFP-BirA*). First, we validated the expression in the stable pools by doxycycline induction of the BirA* fusion proteins, followed by immunoblotting using an anti-FLAG antibody. Figure 1B (left blot) shows that upon induction, bands corresponding to the expected molecular weight of the GFP-BirA* and BirA*-CCP6 fusions can be observed. We further analyzed BirA*-dependent cis- and trans-biotinylation of bait proximal proteins. After doxycycline induction, supplementation of free biotin to the cell culture triggered the biotinylation of cellular proteins, indicating that the promiscuous biotin ligase was active, as evidenced by the distinctive biotinylation pattern observed on Western blots probed with streptavidin-horseradish peroxidase (HRP, Figure 1B, right blot). While the biotinylation pattern observed in Figure 1B is specific for each of the BirA* fusion proteins, biotin induces negligible biotinylation in WT cells by Western blot analysis (i.e., they lack the expression of any BirA* fusion protein).

Further, we analyzed the correct subcellular localization of the BirA*-CCP6 fusion protein by means of fluorescence microscopy using the endogenous protein as a reference. As observed in Figure 1C, anti-FLAG-stained cells expressing FLAG-BirA*-CCP6 displayed a similar localization pattern as the immunostaining observed for native CCP6. This pattern is consistent with the localization pattern previously described for CCP6, with immunoreactivity predominantly localized in the cytoplasm besides structures resembling the Golgi apparatus as well as the centrioles [21,22,26].

CCP6 and GFP proximal proteins were isolated and analyzed using a modified BioID workflow as recently reported in [38]. Biotinylated proteins from the cell pool’s lysate were isolated using streptavidin agarose, trypsin-digested, and the resulting peptides subjected to LC-MS/MS analysis. Three biological replicates per setup were analyzed, leading to the identification of a total of 1550 unique proteins. These proteins were quantified using label-free quantification (LFQ) intensities. This analysis allowed us to distinguish specific from non-specific binders and led to the identification of a total of 37 candidate interactors significantly enriched in our CCP6 BioID dataset when compared with the GFP dataset (volcano plot depicted in Figure 2 and Appendix A).

Expectedly, CCP6 was found as highly enriched, indicative of extensive cis-biotinylation activity of the BirA*-CCP6 fusion protein, typical for BioID experiments. Further, CCP6 proximal partners included the nonsense-mediated mRNA decay factor SMG7, a protein previously reported to interact physically with CCP6 [39]. In fact, SMG7 is also known to physically interact with other proteins enriched in the CCP6 BioID setup, such as PCM1, CLSPN, SSX2IP, and OFD1 [40,41,42]. 

The list of CCP6 proximal proteins resulting from our BioID dataset was consistent with previous studies reporting on the possible centriolar and cilia-related roles of CCP6 [26], as significantly enriched GO terms (Table 1) were linked to cilium assembly (GO: 0060271 and GO:1905515) and centrosome/centriole-related processes (GO:0071539 and GO:0007099). Among these enriched GO clusters, proteins directly related to ‘cilium assembly’ (GO:0060271) and ‘non-motile cilium assembly’ (GO:1905515) were NUDCD3, PIBF1, CEP131, PCM1, C2CD3, CEP350, CCDC66, SSX2IP, KIAA0753 and OFD1 (Figure 3 purple and red and Appendix A and Table 1). Interestingly, five of the potential interactors (PIBF1, CSPP1, OFD1, C2CD3, and KIF7) have been linked to a single ciliopathy, the Joubert syndrome [43,44,45,46], reinforcing the role of CCP6 activity in cilia regulation. The list of CCP6 enriched proteins that were related to ‘protein localization to the centrosome’ (GO:0071539) and ‘centriole replication’ (GO:0007099) included PIBF1, CEP131, KIAA0753, NUDCD3, CEP350, CEP192, C2CD3, PCM1, CCP110, CEP152 and CCDC14 (Figure 3 orange and red and Appendix A and Table 1). In order to further investigate the subcellular localization of CCP6, we analyzed our BioID data against the entire database of the Human Cell Map [40]. Among the 10 most similar baits found in the Human Cell Map (Table 2 and Appendix A), 7 of them are predicted to have a centrosomal localization according to the Human Cell Map, further linking CCP6 to the centrosome [40].

In addition, we found the polyglutamylase TTLL5 as a putative CCP6 interactor in our BioID dataset. This finding suggests that CCP6 and TTLL5 could work together in tandem for regulating optimal polyglutamylation levels. TTLL5 has been found as prey in other BioID experiments present in the Human Cell Map (https://cell-map.org/, accessed on 12 December 2022) [40] and also performed in HEK 293 cells. In four cases the baits were centrosomal proteins (i.e., PCM1, CEP135, CCDC14, and NDC80) and in two other the baits were keratins (i.e., KRT19 and KRT8) with localization at intermediate filaments. In this context, it is worth noting that PCM1 and CCDC14 appear in the candidate list of CCP6 interactors. TTLL5 and TTLL12 are the only two TTLLs found as prey in the Human Cell Map. Similarly, a network of centriolar satellite proteins built with proximity labeling contains the tubulin polyglutamylases TTLL1 and TTLL5 [47]. Further, according to the Human Protein Atlas (https://www.proteinatlas.org/, accessed on 30 December 2022) [48], six TTLLs are expressed in HEK 293 cells, including TTLL1, TTLL3, TTLL4, TTLL5, TTLL7, TTLL11, and TTLL12 (Appendix A). TTLL12, TTLL3, and TTLL4 are the TTLLs more abundantly expressed in HEK 293 cells (Appendix A). From these enzymes, TTLL1 and TTLL11 were shown to localize to the basal body, whereas TTLL4, TTLL5, and TTLL7 are localized to both the cilia shaft and the basal body [49].

Our dataset also showed the GO term ‘snoRNA localization’ (GO:0048254) as significantly enriched (Figure 3 green and Appendix A and Table 1), with two of the CCP6 enriched proteins (PIH1D1 and ZNHIT6) being involved in this specialized cellular process. However, the relevance of this association between CCP6 and these RNA-related processes must be further investigated.

### 2.2. CCP6 Interacts with PCM1 and PIBF1

Following up on the proteomic analysis of the CCP6 proximal environment, we focused on the validation of some pericentriolar matrix proteins as putative CCP6 interactors. Specifically, we wanted to gain insight into the association of CCP6 with cilium assembly and protein localization to the centrosome. Hence, we chose the Pericentriolar material 1 protein PCM1 and the Progesterone-induced-blocking factor 1 PIBF1, another pericentriolar protein, as starting points for our validation efforts.

As observed in Table 1, we found that PCM1 and PIBF1 were associated with both the ‘non-motile cilium assembly’ and ‘protein localization to the centrosome’ GO terms. PCM1 is one of the best-characterized components of the centrosome assembly. This protein is known to be required for anchoring microtubules to the centrosome and is involved in cilia biogenesis [50,51,52]. PIBF1 (also known as CEP90) is another well-characterized component of the centriolar material and, in association with PCM1, is predicted to be involved in primary cilia formation and maintenance of mitotic spindle integrity [53,54].

First, we assessed the potential colocalization of PCM1 and PIBF1 with CCP6 in HEK 293T cells using high-resolution confocal microscopy. We observed that PCM1 shows a clear centriolar localization pattern (Figure 4A, red), in agreement with reports in the literature [42]. In line with this finding, PCM1 and CCP6 (Figure 4A, green) displayed a colocalization pattern limited to the centriolar region. In contrast, for CCP6 (Figure 4B, green) and PIBF1 (Figure 4B, red) a broader colocalization pattern is observed, seemingly colocalizing in different cytoplasmic structures besides the centriolar region.

To further investigate the physical interaction of CCP6 with PCM1 or PIBF1, we performed co-immunoprecipitation (co-IP) assays using a specific anti-HA antibody in cells overexpressing HA-tagged CCP6. However, CCP6 did not co-immunoprecipitate PCM1 or PIBF1 (Appendix A), suggesting that the interaction between CCP6 and PCM1 or PIBF1, if proven to be true, may be transient or weak. Thus, we performed a proximity ligation assay (PLA), an approach previously used to validate BioID screenings and with the potential to assess the presence and location of weak binary interactions that, otherwise, could not be detected by harsher methods such as co-immunoprecipitation [55,56,57]. In PLA, the two proteins under evaluation are targeted by primary antibodies raised in different species, and specific DNA probes are linked to the corresponding secondary antibodies (i.e., PLA probes), which will only be able to hybridize provided both proteins are in proximity (Figure 4E). The in-situ amplification of the DNA probes using rolling circle amplification in combination with fluorescent nucleotides subsequently leads to the visualization of the interaction between the proteins of interest as an intense fluorescent (red) dot with high-resolution microscopy [58]. As can be seen in Figure 4C, PLA showed a positive interaction between CCP6 and PCM1, with cytoplasmic localization of the complex in a structure that resembles the centriole. Therefore, PLA confirms our BioID results, revealing the pericentriolar matrix protein PCM1 as a physical direct interactor of CCP6. However, this interaction does not seem to be restricted to the cellular centriole, since we also observed less intense positive PLA signals at other cellular locations throughout the cytoplasm (Figure 4C). The presence of alternative interacting locations apart from the centrosome could be due to the recruiting capacity of PCM1 [59], which could be involved in CCP6 localization to the centriole.

The PLA assay was also used to evaluate the CCP6-PIBF1 interaction in HEK 293T cells. As observed in Figure 4D, the PLA assay showed a positive interaction between CCP6 and PIBF1. Strikingly, PLA displayed a singular intense signal with a specific localization in cells which, again, resembles the centrosome, whereas both proteins seemed to colocalize throughout the entire cytoplasm based on the immunofluorescence analysis (Figure 4B), which could indicate a potential increase in CCP6/PIBF1 affinity when localized centrosomally.

After demonstrating the physical interaction between PIBF1-CCP6 and between PCM1-CCP6, we wanted to evaluate whether these proteins are substrates of CCP6 enzymatic activity. To this end, we pulled down PCM1 and PIBF1 using their specific antibodies and probed these proteins with the GT335 monoclonal antibody, directed against glutamate side chains. However, we were not able to observe polyglutamylation signals for any of these proteins in Western blots (Appendix A). It is worth noting that the GT335 antibody is known to recognize polyglutamylation when this PTM is located in an acidic environment. Thus, further experiments are needed, using either different anti-polyglutamylation antibodies, or alternative approaches such as mass-spectrometry, in order to fully discard the presence of polyglutamate side chains in PIBF1 or PCM1.

### 2.3. CCP6 Interacts with NudC

We then focused on NudC, a BioID-informed candidate interactor of CCP6 that is not known to display a centriolar localization. NudC plays a role in neurogenesis and neuronal migration [60] and is required for the proper formation of mitotic spindles and chromosome separation during mitosis, as well as for cytokinesis and cell proliferation [61,62].

To characterize the physical interaction of CCP6 and NudC, we first assessed the subcellular localization of NudC in HEK 293T cells by high-resolution confocal microscopy. We observed that NudC (Figure 5A, red) shows general cytoplasmatic staining, where it has been previously described to interact with tubulin and dynein [63]. Further, both CCP6 (Figure 5A, green) and NudC exhibited co-localization in the cytoplasm. To further investigate the physical interaction of CCP6 with NudC, we performed a co-IP analysis with a similar strategy as described for PIBF1 and PCM1 in the previous section. We observed that CCP6 did not co-immunoprecipitate NudC (Appendix A), suggesting that, if present, the physical interaction between CCP6 and NudC might be transient or weak. Nevertheless, by means of PLA, we confirmed a positive interaction between CCP6 and NudC (Figure 5B). PLA signal suggested a cytoplasmic localization in the perinuclear region for the CCP6-NudC complex, where it presents a predominantly diffuse signal around the nucleus, with occasional punctuate patterns also around the nucleus. Thus, these results confirm NudC as a physical interactor of CCP6.

To gain further insights into the association between CCP6 and NudC, we then focused on the role of NudC in cytokinesis. NudC is known to regulate intercellular bridge elongation [61], a process that is not well understood yet. As observed in Figure 5C, localization of both CCP6 (green) and NudC (red) is highly concentrated in the midbody of dividing cells. Moreover, these two proteins display a clear co-localization pattern in the midbody, which could suggest a role of CCP6 in the correct regulation of cytokinesis. It would be interesting to assess whether glutamylation plays a role in the fine modulation of cytokinesis by NudC, as has been already demonstrated for other PTMs [61,62]. Finally, we were not able to detect polyglutamylation side chains in NudC when this protein was probed with the GT335 antibody by Western blot analysis (Appendix A).

## 3. Discussion

In the present study, we have identified 37 potential functional partners of CCP6 using the BioID methodology, thereby gaining insight into the role of CCP6 in a cellular context. Among the identified partners, at least 14 proteins were associated with the centrosome, centriole, cilia biogenesis, and Joubert syndrome ciliopathy (Figure 2 and Figure 3, Table 1 and Appendix A) [37,46,51,64]. Notably, one of the proteins in the candidate list (SMG7) is a known interacting partner of CCP6 [39]. By means of in-situ PLA, our work was able to show that three putative interactors identified by BioID (i.e., PCM1, PIBF1, and NudC) represent true physical interactors of CCP6. Overall, our data suggest that the list of 37 candidate proteins contains a high proportion of bona fide CCP6 interacting partners. To the best of our knowledge, this is the first research article studying the protein interactors of a cytosolic carboxypeptidase.

Our results support the hypothesis for a role of CCP6 (and other CCPs) as a tubulin deglutamylase associated with cilia formation and maintenance [26]. Functional and localization studies have shown that CCPs are mainly expressed in cilia-containing tissues in different organisms (e.g., CCP1 is enriched in the olfactory bulb and the cerebellum, while other CCPs are also found in the testis, trachea, lungs, kidneys, and eyes) [65,66,67]. At the cellular level, axonemes of cilia and flagella, as well as centrioles of mammalian centrosomes, are cellular structures known to harbor high polyglutamylation levels [68,69]. Further, polyglutamylation of these specific structures is known to modulate the interaction between microtubules and different microtubule-associated proteins (MAPs) such as dynein motors, leading to precise regulation of processes such as axogenesis, stability of the centrosome, cell division, and the sensory and motor functionality of cilia and flagella [70,71,72,73]. Consistent with these data, at the subcellular level, CCPs are found to co-localize with γ-tubulin from the centrioles, in all phases of the cell cycle, and with polyglutamylated tubulin from the basal bodies in ciliated cells [26]. Our BioID results not only confirm a CCP6 localization at centrioles and centrosomes but also highlight two GOBP terms enriched in the CCP6 dataset that are related to cilia assembly: ‘Cilium assembly (GO:0060271)’ and ‘Non-motile cilium assembly (GO:1905515)’.

The relevance of CCP’s role in ciliated tissues has also been inferred from the abnormalities observed in CCP knockouts, since both CCP1 and CCP5 deficiency lead to cilia dysfunction in different organisms. For instance, studies in *Caenorhabditis elegans* showed that a single amino acid substitution in the conserved N-domain of CCPP-1 (a CCP1 homolog) causes the accumulation of the motor protein kinesin KLP-6, depriving the cell of cilia-based signal transduction. In turn, this leads to progressive deterioration of the ciliary axoneme, suggesting that CCP activity is necessary for the maintenance of the structural integrity of the cilia [65]. In mice, the CCP1 loss of function leads to the *pcd* (Purkinje cell degeneration) phenotype [16,74]. These mice display several alterations, such as degeneration of Purkinje cells, thalamic neurons, retinal photoreceptors, and olfactory bulb mitral cells, as well as reproductive abnormalities [75,76], all of which affect ciliated tissues. In humans, the loss of function of CCP1 leads to childhood-onset neurodegeneration with cerebellar atrophy (CONDCA), a pathology with a similar phenotype to the ataxic disease in *pcd* mice [18,77]. Correspondingly, CCP5 is the main deglutamylase in the cilia of zebrafish and plays an important role in ciliogenesis. CCP5 knockdown causes curvature of the fish body, cyst formation in the protonephridium ducts, and hydrocephalus due to the enlargement of the cerebral ventricles caused by the paralysis of the cilia of the ependyma and accumulation of cerebrospinal fluid [78,79]. In humans, it has been observed that CCP5 may have a key role in axoneme polyglutamylation, since the regulation of glutamylation levels is a major contributor to cilia signaling and it is necessary for correcting signaling defects in ciliopathies [35]. In this context, the role of CCP6 in cilia-related structures has not been studied in detail. So far, the association of CCP6 and cilia has been inferred from the overexpression of CCP6 in ciliated tissues and its subcellular localization in the basal body [26,78,79]. Mapping of the CCP6 proxeome reinforces the participation of CCP6 in controlling and maintaining ciliary structure and could be used as a starting point to further expand on this knowledge.

Intriguingly, five of the candidate interactors found in our BioID-based interactomic study (i.e., C2CD3, PIBF1, CSPP1, OFD1, and KIF7) are related to a single rare pathology, the Joubert syndrome. This ciliopathy is characterized by cerebellar hypoplasia, and neurological symptoms such as ataxia, psychomotor retardation, and oculomotor apraxia. Moreover, JS is commonly accompanied by a variety of multiorgan signs and symptoms, such as retinal dystrophy, nephronophthisis, liver fibrosis, and polydactyly, all of which are related to ciliopathy illnesses such as Meckel–Gruber Syndrome, and Bardet–Biedl syndrome [80]. CSPP1 and PIBF1 are known to be located at the centriole [81,82], OFD1 in the basal body [83], and KIF7 in the axoneme distal tip [84]. We hypothesize that polyglutamylation could provide the link between CCP6 and JS, since a minimal regulation of polyglutamylation in the ciliary axoneme is needed for cilia stability, while it is essential for signaling [49]. Interestingly, polyglutamylation regulation at the axoneme is mediated by another JS-related protein, ARL13B, which imports TTLL5 and TTLL6 polyglutamylases into the cilia [35]. Further, polyglutamylation was described to affect the JS-related ARMC9/TOGARAM1 protein complex [36], indicating that this PTM is closely related to this ciliopathy. All this suggests that CCP6 may play a role in the development of JS by regulating cilia polyglutamylation. Thus, targeting the polyglutamylation machinery could be evaluated as a potential therapeutic strategy to correct the signaling defects in ciliopathies.

An interesting finding of our study is the presence of TTLL5 as a candidate CCP6 interactor, suggesting a possible coordinated action between these complementary enzymes with opposing actions. In support of the hypothesis that CCPs and TTLLs could work together in tandem for regulating optimal polyglutamylation levels, CCP1 and TTL4 have also been described in the literature to physically interact [85].

In the present work, we have shown that PCM1 is a direct interactor of CCP6. PCM1 has been implicated in centrosomal protein trafficking in a variety of studies, as its cellular localization is not restricted to centrosomes. PCM1 is known to display recruiting capacity [59] and, thus, it could function as a transporter or assembly factory for other centrosomal proteins, such as CCP6 to the pericentriolar matrix [50]. Moreover, we showed the direct interaction between CCP6 and PIBF1, another centriolar protein that physically interacts with PCM1 at centriolar satellites. The PIBF1-PCM1 interaction is essential for the centrosomal accumulation of centriolar satellites and eventually for primary cilia formation [53]. PIBF1 is also required for BBS4 loading into centriolar satellites and its localization in primary cilia. Therefore, further investigation into the interaction between CCP6 and these two proteins could provide interesting insights into the mechanism that allows CCP6 to dynamically localize to centrioles and primary cilia.

In addition, we validated the physical interaction between CCP6 and NudC, which is known to play a role in neurogenesis and neuronal migration [60,63]. Moreover, NudC is required for the correct formation of mitotic spindles and chromosome separation during mitosis [62], and for cytokinesis and cell proliferation [61]. The interaction between CCP6 and NudC supports the idea that CCP6 is involved in other cellular functions besides cilia regulation. NudC function is known to be regulated by post-translational modifications, including deacetylation by HDAC3 [62] and phosphorylation by at least three mitotic kinases (i.e., Cdk, Plk1, and Aurora B) [61]. Hence, it would be interesting to thoroughly assess whether glutamylation would also play a role in the finetuning of NudC functioning in these key cellular processes. 

In summary, this study has uncovered the cellular interactome of CCP6 using BioID and PLA methodologies, two approaches that enable the detection of weak and transient interactions. Our results support the idea that CCP6 is an important cilia-related effector protein, a finding that could be used to evaluate therapeutic strategies targeting this enzyme to correct defects in ciliopathies. Further characterization of CCP6 putative interactors might provide insights into whether CCP6 is an important player in other cellular mechanisms. Finally, it is important to note that these results were obtained using isoform 1 of CCP6 (Q5VU57-1). As it occurs with most of CCPs, different CCP6 isoforms differ in the length of protein termini and internal loops. Thus, it would be interesting to explore in future studies how the difference in the sequence of the different CCP6 isoforms affects their protein–protein interactions.

## 4. Materials and Methods

### 4.1. Cell Lines and Cell Culture

Adherent HEK 293T (ATCC) and Flp-In T-REX 293 cells (Thermo Fisher Scientific, Waltham, MA, USA) cell lines were cultured in DMEM (Gibco, Thermo Fisher Scientific) supplemented with 10% heat-inactivated fetal bovine serum (FBS, Gibco, Thermo Fisher Scientific). Cells were incubated at 37 °C in a humidified atmosphere with 5% CO_2_.

### 4.2. Interactomic Studies

#### 4.2.1. Molecular Cloning and Generation of Stable Cell Lines

All generated plasmids used in this study were constructed by PCR and restriction enzyme-based cloning. BioID scaffold plasmids constructed for inducible expression (pcDNA5_FRT-TO_FLAGBirA-[MCS]) were kindly provided by Dr. Brian Raught (Princess Margaret Cancer Centre, Toronto, ON, Canada) [86]. A human CCP6 construct encoding for the canonical CCP6 form (isoform 1, UniProt ID Q5VU57-1) previously generated in our laboratory [87], served as a template to generate NotI- and XhoI-flanked PCR products to clone into the NotI/XhoI sites of pcDNA5_FRT-TO_FLAGBirA-[MCS]. Primers were designed to fuse CCP6 gene in-frame with an N-terminal BirA*-FLAG tag encoded by the BioID scaffold plasmid, in order to rule out potential protein function interferences due to the presence of the carboxypeptidase catalytic domain at the C-terminal end of CCP6. The construct had 5×glycine-alanine encoding linkers added between BioID and the CCP6 gene to provide flexibility between the modules. All constructs had sequences verified by Sanger sequencing. The pDEST-pcDNA5-BirA-FLAG-GFP control vector was kindly provided by Dr. Anne-Claude Gingras (Lunenfeld-Tanenbaum Research Institute, Toronto, ON, Canada) [88].

Flp-In T-REX 293 cells, which constitutively express the Tet repressor, were used to generate the BioID stable cell pools as follows. Stable cell pools for CCP6 or GFP bait proteins were generated by co-transfection of the CCP6 or GFP expression construct and the Flp recombinase-encoding pOG44 plasmid (Invitrogen, Thermo Fisher Scientific) in a 1:9 ratio (6-well plates at 60–70% confluence) using Lipofectamine 2000 (Invitrogen, Thermo Fisher Scientific) in Opti-MEM according to the manufacturer’s instructions. After recovery from transfection, cells were grown in DMEM containing 10% FBS, 15 µg/mL blasticidin, and 50 µg/mL hygromycin B for selection. Selected colonies were isolated and pooled by trypsinization, collected by centrifugation (1000× *g*, 5 min) at room temperature (RT), and resuspended and maintained in complete DMEM supplemented with 15 μg/mL blasticidin and 50 μg/mL hygromycin B. Cell pools were expanded and grown without tetracycline.

#### 4.2.2. BioID

For performing BioID experiments, low passage cells were plated at 1 × 10^7^ cells/plate in 15 cm dishes. Each replicate was performed with cells with a different passage number and consisted of 6 × 15 cm plates. After 6 h of plating, tetracycline was added to the media at a final concentration of 1 μg/mL. After 24 h, biotin was added to the media to a final concentration of 50 µM, and the cells were incubated for an additional 24 h period. After decanting the media, cells were collected from each plate by pipetting with ice-cold PBS. Cells were centrifuged at 1400 rpm for 5 min and the PBS was decanted. Cells were washed once more with ice-cold PBS before proceeding to cell lysis. Cells were resuspended and lysed in 9 mL stringency buffer (100 mM Tris-HCl, pH 7.5; 150 mM NaCl, 2% (*w*/*v*) SDS, 8 M urea) by gentle rocking for 5–10 min at 4 °C. The cell lysate was sonicated to reduce the viscosity (4 sonication cycles consisting of 3 groups of 25-sec bursts, 40% intensity amplitude, 0.5/0.5 cycles). The sonicated lysates were combined with 500 μL pre-washed streptavidin-conjugated agarose beads (Novagen, Darmstadt, Germany) and incubated overnight at 4 °C with gentle rocking. Bead/lysate mixtures were collected by centrifugation at 500× *g* for 5 min. The beads were then washed 4 times with 1 mL stringency buffer. Washed beads were changed to 1 mL high salt buffer (100 mM Tris pH 7.5; 1 M NaCl) for 30 min at RT with gentle agitation and washed once with 1 mL ultrapure water. Typically, 50 µL of the bead suspension in 1 mL ultrapure water were analyzed by SDS-PAGE after protein elution in elution buffer (2% (*w*/*v*) SDS, 3 mM biotin, 8 M urea in PBS) and following 10 min of heating at 96 °C. Beads were separated from the elution solution by centrifugation. The rest of the washed beads were used for on-bead trypsin digestion.

#### 4.2.3. Mass Spectrometry

##### Sample Preparation

Agarose beads were washed three times with 1 mL of 50 mM ammonium bicarbonate pH 8.0 (ABC buffer) and pelleted by centrifugation at 600× *g* for 2 min. Washed beads were re-suspended in 600 μL ABC buffer, and 1 μg trypsin was added to each peptide mixture. On bead trypsin digestion was incubated overnight at 37 °C with gentle agitation. Next day, additional 0.5 μg of trypsin was added to each sample (in 10 μL 50 mM ABC) and the samples were incubated for an additional 2 h at 37 °C. Trypsinized beads were pelleted (600 g, 2 min) and the tryptic digest was transferred to a low-binding 1.5 mL tube. Beads were then rinsed 2 times with 300 μL of mass spec-grade H_2_O. Rinses were combined with the original supernatant. The pooled fractions were acidified with 10% formic acid (to get a final concentration of 0.2% of formic acid), centrifuged at 16,100× *g* for 20 min, and the supernatant was transferred to a new low-binding 1.5 mL tube and dried in a speed-vac. Samples were resuspended in 25 μL of 2% acetonitrile, 2 mM TCEP buffer (Tris [2-carboxylethyl]phosphine hydrochloride) and centrifuged at 16,100× *g* for 15 min to be further used for analysis.

##### Data Acquisition and Analysis

The BioID peptide samples were introduced into an LC-MS/MS system, an UltiMate 3000 RSLCnano HPLC (Thermo Fischer Scientific) in-line connected to a Q-Exactive instrument (Thermo Fischer Scientific) as previously described [89,90]. The raw data files were processed with MaxQuant [91] using the Andromeda search engine (version 1.5.7.4) [92] and MS/MS spectra searched against the Swiss-Prot database (taxonomy Homo sapiens) extended with the BirA* and GFP sequences. Potential contaminants present in the contaminants.fasta from MaxQuant were also added. A precursor mass tolerance was set to 20 ppm for the first search (used for nonlinear mass recalibration) and set to 4.5 ppm for the main search. The enzyme specificity was defined as trypsin allowing up to two missed cleavages. Variable modifications were set to oxidation of methionine residues and N-terminal protein acetylation. The false discovery rate (FDR) was set to 1% on the identification at both peptide and protein level and the minimum peptide length was set to 7. The minimum score threshold for both modified and unmodified peptides was set to 40. Only proteins that were identified with at least three peptide spectra matches were retained. The match between runs function was enabled and proteins were quantified by the MaxLFQ algorithm integrated into the MaxQuant software (version 1.6.0.1) [93]. Here, a minimum of two ratio counts and only unique peptides were considered for protein quantification. 

For basic data handling, normalization, statistics, and annotation enrichment analysis we used the freely available open-source bioinformatics platform Perseus (version 1.6.15.0) [94] and R (version 4.1.3). Data analysis after uploading the protein groups file obtained from MaxQuant database searching was performed as described previously [89]. The protein list was filtered out to remove contaminants (i.e., proteins present in the contaminants.fasta from MaxQuant), reversed proteins (i.e., proteins that are present in the decoy database used to adjust the identification FDR to 1%), and proteins only identified by site (i.e., proteins that are only identified by peptides carrying peptide modifications, which are often less reliable identifications). Following a standard MaxQuant-Perseus workflow [94], all replicate samples were grouped and LFQ intensities were log2 transformed. Afterward, proteins with less than 2 valid values in at least one group were removed. Assuming the presence of missing data due to the limit of detection, missing values were imputed from a normal distribution around the detection limit (with 0.3 spread and 1.8 down-shift) [94,95]. Afterward, the data were subjected to a scaled median absolute deviation (SMAD) normalization with the R package “marray” (version 1.60.0). Then, a *t*-test (FDR = 0.05, S0 = 0.1) was performed between the GFP and CCP6 groups to generate a volcano plot and detect enrichments in the different BioID setups.

#### 4.2.4. Gene Ontology Analysis

The PPIs identified were subjected to gene ontology analysis using the PANTHER software (version 17.0, http://www.pantherdb.org/, accessed on 12 December 2022) [96,97] for the ‘Biological process complete’ category using the PANTHER overrepresentation test (release 20221013) against the *Homo sapiens* gene database (Fisher’s exact test corrected by False Discovery Rate). The STRING network (version 11.5, https://string-db.org/, accessed on 12 December 2022) [98] was used to further interrogate the identified proteins, and to show any previously known associations in a comparison with the entire STRING database. For this purpose, a minimum interaction confidence score of 0.4 was used. Results from both analyses were combined and plotted using the NORMA web tool (version 2.0) (http://bib.fleming.gr:3838/NORMA, accessed on 12 December 2022) [99]. The identified PPIs were also analyzed using the GOnet web tool (version 1.0) (https://tools.dice-database.org/GOnet/, accessed on 12 December 2022) [100] to show relationships between terms and genes/proteins using the ‘Biological Process’ GO term enrichment analysis in a comparison with all annotated human genes and a minimum q-value threshold of 0.05.

#### 4.2.5. Human Cell Map Analysis

For the analysis of the BioID data in the Human Cell Map [40], we first analyzed the probability of interaction between CCP6 and its preys using SAINTexpress within the CRAPome repository [101,102]. We performed this by using the LFQ values as abundance measure in each of the samples, the GFP-BirA* samples as controls, and using the standard SAINT options. Afterward, the output of SAINTexpress was used as input for the analysis in the Human Cell Map (https://cell-map.org/query/, accessed on 12 December 2022). For this analysis, the AvgIntensity was used as the abundance measure, the BFDR as the confidence score, and a Cutoff of 0.01 in a comparison with the entire database (v1). For the heatmap visualization of the Human Cell Map output, we used the R package “ggplot2” (version 3.4.0) [103].

### 4.3. Co-Immunoprecipitation

HEK 293T cells were seeded at a density of 2 × 10^6^ cells per 100 mm dish. Next day, cells were transiently transfected with linear polyethyleneimine (PEI, 25,000 Da, Polysciences, Warrington, PA, USA) in a 1:3 (*w*/*w*) DNA:PEI ratio using an HA-tagged CCP6 construct in a pTriEX-6 scaffold previously generated in our research group [87]. Cells were exposed to the transfection complex at 1 µg DNA per ml of culture for 24 h. Afterward, cells were collected by trypsinization and centrifugation at 600× *g* for 5 min at RT. Harvested cells were lysed at 4 °C for 1 h with NP40 lysis buffer (0.1% NP40 in PBS) supplemented with protease inhibitors without EDTA (Complete Protease Inhibitor Cocktail EDTA-free, Roche, Basel, Switzerland) and centrifuged at 12,000 rpm for 10 min. Protein G magnetic particles (Dynabeads^®^ Protein G, Thermo Fisher Scientific) were first conjugated with the anti-HA antibody (Thermo Fisher Scientific, 1 μg of antibody in 1.5 mg of magnetic particles) by rotation in a roller for 10 min at RT. Cell extracts were added (1 mL) to the magnetic beads and rotated for 30 min at 4 °C. Magnetic bead-Ab-Ag complex was washed 3 times using NP40 lysis buffer and samples were eluted with 1X SDS loading buffer and boiled for 5 min. The samples were then used for Western blotting analysis. 

### 4.4. Western Blotting

Lysates and eluates were run on 4–12% polyacrylamide gels and transferred to PVDF (Immobilon-P, EMD Millipore, Burlington, MA, USA) for 1.5 h at 100 V constant voltage. Blots were blocked for 30 min with PBST (phosphate buffer saline, 0.05% Tween, Thermo Fisher Scientific) 5% dry non-fat milk (*w*/*v*) and immunoblotted with appropriate antibodies. All antibodies were diluted in PBST 2% milk (*w*/*v*). Primary antibodies were incubated overnight at 4 °C, while secondary antibodies were incubated for 1 h at RT. Antibodies used were anti-FLAG (Sigma-Aldrich, St. Louis, MO, USA, 1:5000 dilution), anti-Streptavidin (Thermo Fisher Scientific, 1:1000 dilution), anti-HA (1:5000, Thermo Fisher Scientific) anti-GAPDH (Thermo Fisher Scientific 1:10,000 dilution), anti-PCM1 (Santa Cruz Biotechnology, Dallas, TX, USA, 1:500 dilution), anti-PIBF1 (Santa Cruz Biotechnology, 1:500 dilution), anti-NudC (Santa Cruz Biotechnology, 1:500 dilution), anti-GT335 (Adipogen, San Diego, CA, USA, 1:5000 dilution), goat anti-rabbit HRP (Bio-Rad, Hercules, CA, USA, 1:5000 dilution) and goat anti-mouse HRP (Bio-Rad, 1:5000 dilution). Western blots were visualized with Luminata Forte (Merck Millipore, Darmstadt, Germany) and a VersaDoc imaging system (Bio-Rad). Image intensity histograms were adjusted, and images were analyzed with ImageLab software (version 6.1.0 build 7, Bio-Rad).

### 4.5. Confocal Microscopy

#### 4.5.1. Proximity Ligation Assay (PLA)

For CCP6 interactor validation we performed a Duolink^®^ Proximity Ligation Assay (PLA, Sigma Aldrich) which allows in situ evaluation of protein interactions that can be readily detected and localized in unmodified cells and tissues. Briefly, 4% paraformaldehyde fixed cells samples were blocked with Duolink^®^ Blocking Solution for 60 min at 37 °C. Blocked samples were then incubated with specific primary antibodies overnight at 4 °C with gentle agitation. Secondary antibodies conjugated with oligonucleotides (PLUS and MINUS probes) were added to the reaction and incubated for 1 h at 37 °C. Ligation solution, consisting of two oligonucleotides and ligase, was added, and incubated for 30 min at 37 °C. In this assay, the oligonucleotides hybridize to the two PLA probes and join to a closed loop if they are in close proximity. Amplification solution, consisting of nucleotides and fluorescently labeled oligonucleotides, was added together with polymerase, and incubated for 100 min at 37 °C. The oligonucleotide arm of one of the PLA probes acts as a primer for “rolling-circle amplification” using the ligated circle as a template, and this generates a concatemeric product. Fluorescently labeled oligonucleotides hybridize to the amplified product. The PLA signal was visible as a distinct fluorescent spot and was analyzed by confocal microscopy (TCP SP5, Leica, Wetzlar, Germany). All proximity ligation assays were performed in three independent experiments with a minimum of 75 total cells. Positive interaction was observed in 73 ± 4% (PCM1), 37 ± 6% (PIBF1) and 52 ± 10% (NudC) of the cells. Control experiments included routine immunofluorescence staining of proteins of interest under identical experimental conditions. 

#### 4.5.2. Immunocytochemistry and Image Acquisition

Cells were fixed with 4% paraformaldehyde for 30 min at RT. Cells were further permeabilized with PBS containing 0.5% Triton X-100 for 30 min, blocked with 1% BSA in PBS 0.05% Tween for 30 min and then incubated with the appropriated primary antibodies overnight at 4 °C. For detection of the BioID constructs, anti-CCP6 (anti-agbl4, Abcam, Cambridge, UK, 1:100 dilution) and anti-FLAG (Thermo Fisher Scientific, 1:1000 dilution) were used. At least three independent experiments were performed for each condition and a minimum of 100 cells in total were analyzed for assessing the protein subcellular localization. All the analyzed cells stably expressed the respective BirA* constructs. For localization studies, anti-CCP6 (anti-agbl4, Abcam, 1:50 dilution), anti-PCM1 (Santa Cruz Biotechnology, 1:100 dilution), anti-PIBF1 (Santa Cruz Biotechnology, 1:100 dilution) and anti-NudC (Santa Cruz Biotechnology, 1:100 dilution) were used. Cells were then washed and incubated with Alexa anti-rabbit 488 and Alexa anti-mouse 555 (Life Technologies, Carlsbad, CA, USA) for 1 h at RT and nuclei were further counterstained with DAPI. Images for localization were obtained from at least three independent experiments. All images are representative of the obtained data. Confocal images were obtained on a TCS SP5 (Leica) confocal microscope using a 63×/1.4–0.6 PL APO oil immersion objective, ∼1.2 airy unit pinhole aperture and appropriate filter combinations. Images were acquired with 405 nm Diode laser, 488 nm multiline Argon laser and 594 nm Helium-Neon lasers. 

## Figures and Tables

**Figure 1 ijms-24-01273-f001:**
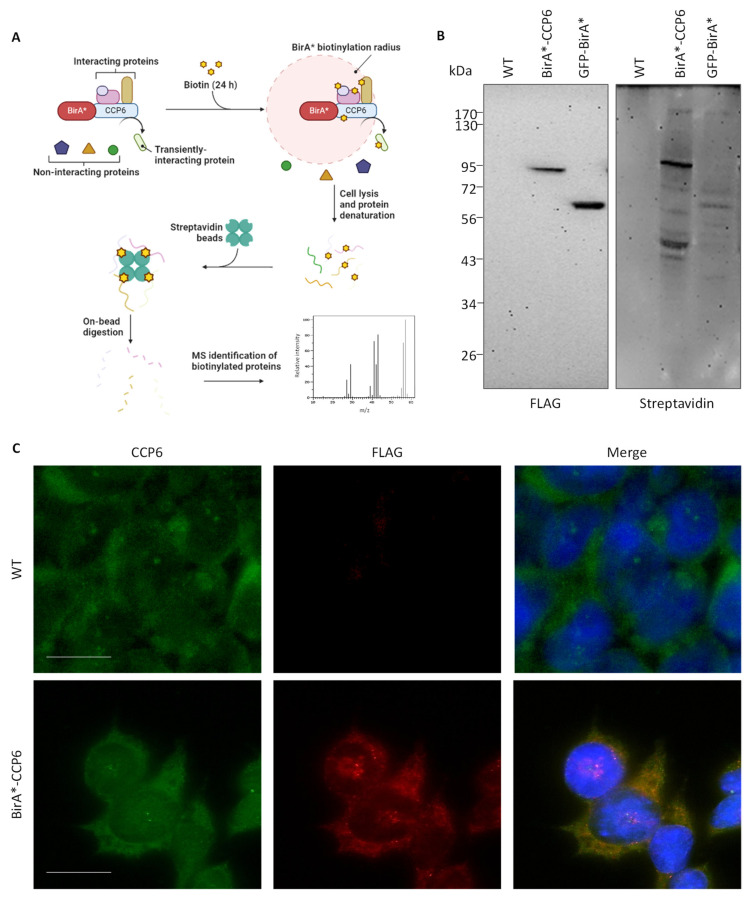
BioID cell pool generation and validation. (**A**) Schematic diagram of the BioID workflow for BirA*-CCP6. (**B**) Western blot confirmation of BirA*-CCP6 and GFP-BirA* expression using anti-FLAG antibody, (left), and the biotinylating activity of the expressed constructs using streptavidin-HRP, (right). (**C**) Immunocytochemistry of the BirA*-CCP6 construct and endogenous CCP6 in HEK 293 cells. Scale bar = 10 µm.

**Figure 2 ijms-24-01273-f002:**
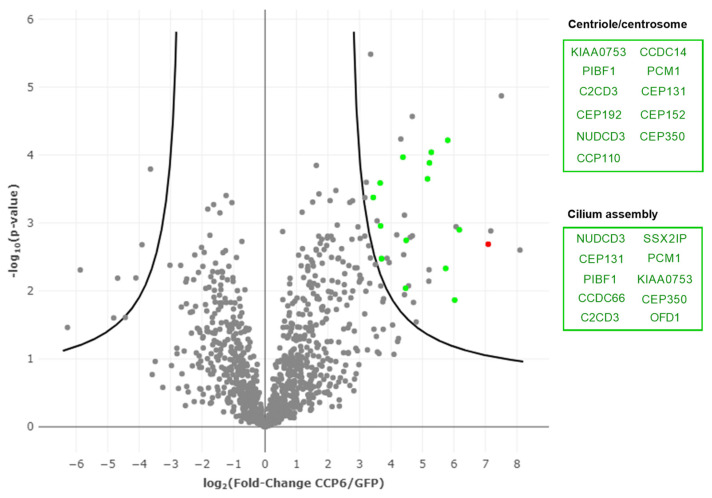
BioID volcano plot of CCP6/GFP proxeome in HEK 293 cells. Volcano plot displaying the statistical significance (*p*-value) versus the fold enrichment of proteins identified using BirA*-CCP6 when compared to the GFP-BirA* control experiment. Among the enriched proteins in the CCP6 setup we found CCP6 (red dot), as well as candidate interactors. Some of the CCP6 candidate interactors are related to centrioles and centrosomes (GO:0071539 and GO:0007099) and cilium assembly (GO:0060271 and GO:1905515) (green dots). Boxes represent the corresponding genes of enriched gene ontology processes.

**Figure 3 ijms-24-01273-f003:**
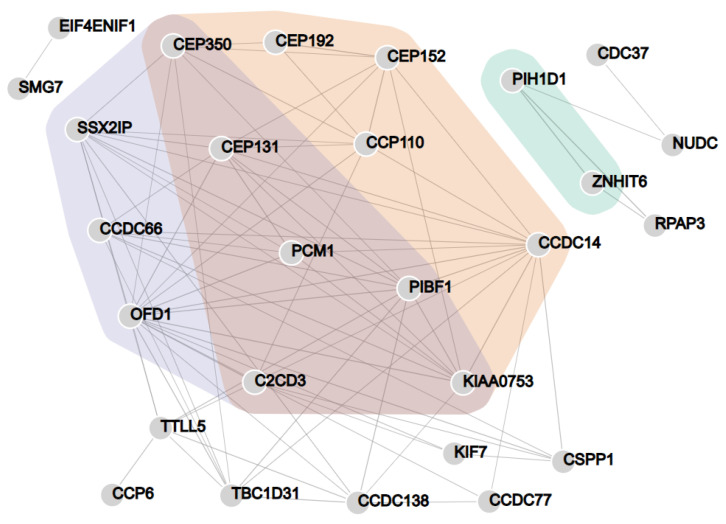
Network analysis of CCP6 proxeome in HEK 293 cells. Interaction data were retrieved from the STRING database for CCP6 proximal proteins and plotted with the NORMA web tool (version 2.0) for enriched GO terms. Proteins related to cilium assembly (cilium assembly [GO:0060271] and non-motile cilium assembly [GO:1905515]; purple and red), centriole and centrosome-related processes (protein localization to the centrosome [GO:0071539] and centriole replication [GO:0007099]; orange and red), and snoRNA localization (green) are highlighted (see also Table 1).

**Figure 4 ijms-24-01273-f004:**
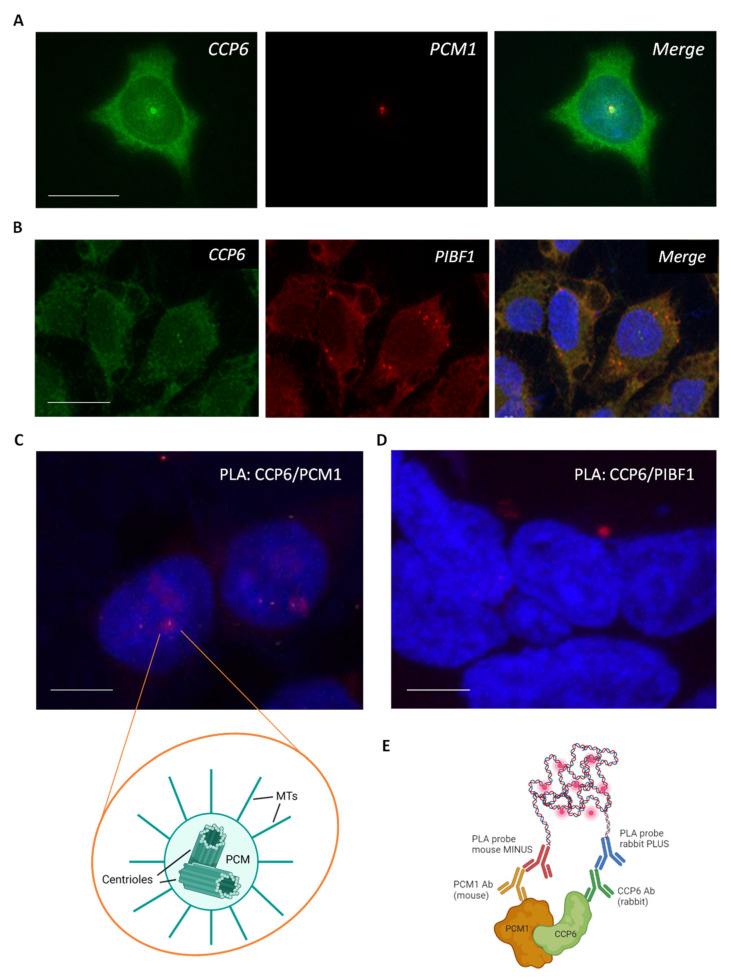
Characterization of CCP6-PCM1 and CCP6-PIBF1 interaction. (**A**) Immunocytochemistry of endogenously expressed CCP6 (green) and PCM1 (red) indicative of their intracellular localization. Scale bar: 10 µm. (**B**) Immunocytochemistry of CCP6 (green) and PIBF1 (red) showing their intracellular localization. Scale bar: 10 µm. (**C**) Representative PLA signal of CCP6-PCM1 complex. Positive PLA signal was observed in 73 ± 4% of the cells. The scheme shows the general structure of a centrosome. MTs: microtubules. PCM: pericentriolar matrix. Scale bar: 5 µm. (**D**) Representative PLA signal of CCP6-PIBF1 complex. Positive PLA signal was observed in 37 ± 6% of the cells. Scale bar: 5 µm. (**E**) Schematic representation of the PLA assay confirming the CCP6-PCM1 interaction. All experiments were performed in HEK 293T cells. Nuclei were stained with DAPI (blue) in all conditions. The interaction in PLA assays is represented by fluorescent rolling circle products (red dots). All images were obtained by immunodetection by means of confocal microscopy.

**Figure 5 ijms-24-01273-f005:**
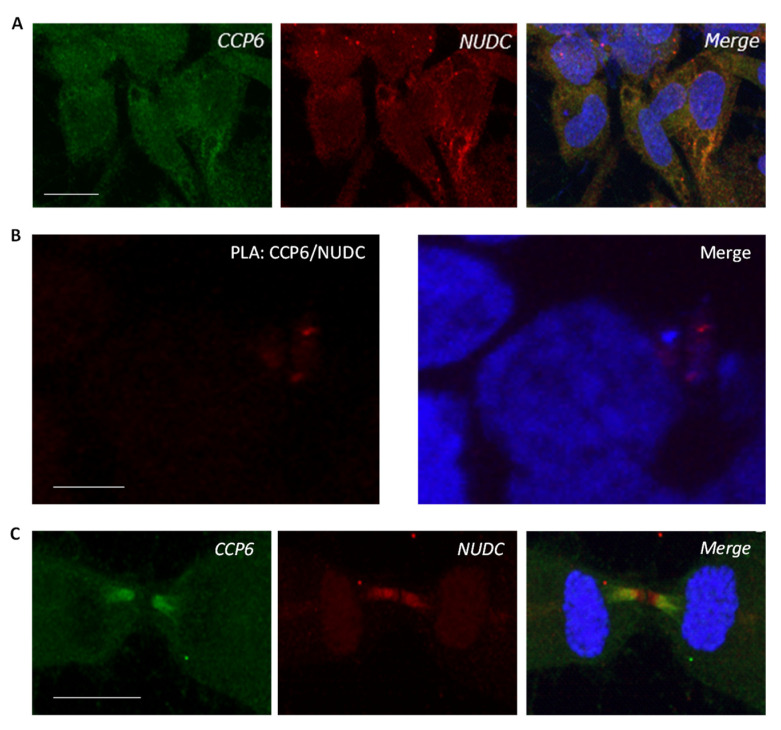
Characterization of the CCP6-NudC interaction. (**A**) Immunocytochemistry of endogenously expressed CCP6 (green) and NudC (red) indicative of their intracellular localization in HEK 293T cells by immunodetection by means of confocal microscopy. Scale bar: 10 µm. (**B**) Representative PLA signal of CCP6-NudC complex in HEK 293T cells. The interaction is represented by fluorescent rolling circle products (red dots). Positive PLA signal was observed in 52 ± 10% of the cells. Scale bar: 5 µm. (**C**) Immunocytochemistry of endogenous CCP6 (green) and NudC (red) indicative of their localization in the midbody of cells in cytokinesis. Scale bar: 10 µm. Nuclei were stained with DAPI (blue) in all cases.

**Table 1 ijms-24-01273-t001:** Enriched Gene Ontology Biological Processes (GOBP) for the CCP6 proximal interactome. The significant CCP6-specific functions were determined by GOBP enrichment analysis (*p*-value < 0.05).

Gene Ontology Biological Processes	Count	*p*-Value	Identified Proteins
Cilium assembly (GO: 0060271)	10	3.23 × 10^−10^	NUDCD3, PIBF1, CEP131, PCM1, C2CD3, KIAA0753, CEP350, CCDC66, SSX2IP, and OFD1
Protein localization to the centrosome (GO:0071539)	9	2.39 × 10^−18^	PIBF1, CEP131, KIAA0753, NUDCD3, CEP350, CEP192, C2CD3, PCM1, and CCDC14
Centriole replication (GO:0007099)	5	1.54 × 10^−9^	CCP110, CEP192, CEP152, KIAA0753 and C2CD3
Non-motile cilium assembly (GO:1905515)	5	9.68 × 10^−8^	PCM1, C2CD3, PIBF1, CEP131 and CEP350
snoRNA localization (GO:0048254)	2	6.90 × 10^−5^	PIH1D1 and ZNHIT6

**Table 2 ijms-24-01273-t002:** Expected cellular locations of the ten most similar baits found in the Human Cell Map as determined by Jaccard distance calculation [40].

Cellular Localization ^1^	Count	Identified Baits ^2^
Centrosome	7	SASS6 (0.832), STIL (0.843), PCNT (0.846), CCDC14 (0.851), LATS1 (0.856), CEP135 (0.862), CDK5RAP2 (0.875)
Cell junction	2	LATS1 (0.856), AMOT (0.860)
Golgi membrane	1	GOLGA1 (0.874)
Intermediate filament	1	KRT19 (0.868)

^1^ Expected cellular localizations according to the Human Cell Map (https://cell-map.org/, accessed on 12 December 2022). ^2^ In parentheses, the Jaccard distances to the CCP6 setup are indicated.

## Data Availability

The authors declare that all the data presented in this study are available in the paper or in the Appendix A. Additional raw data supporting the findings of this study are available from the corresponding author upon request.

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
