# Peer review of "Proximity Mapping of CCP6 Reveals Its Association with Centrosome Organization and Cilium Assembly"

_ijms, 2023, doi:10.3390/ijms24021273_

Round 1
Reviewer 1 Report
This manuscript describes a set of elegant studies to identify proteins that colocalize with the enzyme CCP6, which removes polyglutamate side chains from tubulin and other cellular proteins. CCP6 is a relatively understudied protein, and the present manuscript greatly contributes to the field. Specifically, the identification of proteins that associate with CCP6 is helpful towards understanding the function of this enzyme. The manuscript is very well written. Supplementary table S1 is nicely presented and should be easily understood by interested readers. There are some minor issues that should be resolved:
1) Question: How many interacting proteins really were identified? The abstract and text say 41 interacting proteins identified, and Table S1 is referred to in the text. However, several of these hits were identified by only 1 or 2 peptides, and often in proteomics there is a cutoff of 3 or more peptides per protein. It is highly recommended that 'hits' with only 1 or 2 peptides are removed from the table. It won't impact on the overall discussion, as the proteins identified from 1 or 2 peptides are not discussed further in the paper.
2) A related issue is the total number of proteins identified (including those not specifically associated with CCP6). Text (lines 126-137) mentions 1848 total proteins, but as mentioned in point #1, many of these were from only a single peptide identification (if I’m correctly interpreting Table S1, specifically column X). Of the total list of 1848 proteins, 385 of these were found by only 1 peptide. 313 only found by 2 peptides. Leaving 1150 if one uses a standard cut-off of 3 or more peptides per protein.
3) More details are needed regarding the specific isoform of CCP6 used in the study. The text says that a human CCP6 construct previously generated in the laboratory was used, and cite reference 82. But reference 82 appears to be a thesis, and is not easy to find on-line. The authors should clearly state the form of CCP6 used in the present study because it exists in multiple isoforms (as do most of the CCPs). In the first publications on CCP6 (cited as references 22 and 23 in the manuscript), mouse CCP6 was reported to have two major isoforms that differ in the C-terminal sequence due to alternative splice sites in exon 13. A quick search of the NCBI database shows multiple forms of human CCP6, also differing in the C-term region due to alternative splice sites. The functions of these multiple isoforms is not known. For other proteins, different splice forms can have different protein interactions, and in the future it would be interesting to explore this. No further experiments are necessary at this point, but the authors should at least be clear as to which CCP6 splice form was used in the present study, and comment on the existence of additional CCP6 splice forms, emphasizing that the present results apply to a particular form of CCP6 and it is not known if this is true of the other splice form(s) of CCP6.
Reviewer 2 Report
Polyglutamylation is a post-translational modification of proteins. Our understanding of polyglutamylation is primarily based on its regulation of microtubule function; however, other proteins are modified in this way as well. In order to regulate this modification, deglutamylases are necessary in addition to the polyglutamylases. CCP6 is a deglutamylase enzyme, with important roles in the nervous system and other tissues. The goal of this study by Rodriguez-Calada et al was to identify substrates and interactors of CCP6 in order to understand its cellular function.
The authors use the BioID method of proximity labeling to identify 41 putative interactors of CCP6 in HEK293 cells. Since it has been shown that CCP6 can deglutamylate substrates unrelated to microtubules, this unbiased approach was taken to better understand the possible roles of this enzyme. A large number of these interactors have some role in cilia or centriole function. The authors confirm three of these putative interactors, PCM1, PIBF1, and NudC, and further characterize them using immunocytochemistry and the proximity ligation assay. These results seem to continue to support a central role for CCP6 in the modification of microtubule-based structures; two of the three (PCM1 and NUDC) have clear associations with microtubule-based structures.
The manuscript was well-written and experiments designed well. It adds nicely to the literature that largely shows the CCPs to be involved in modifications of microtubule structures. While most of my suggestions are minor, one suggestion of a larger scope might be made:
The authors use extensive use of fluorescence microscopy throughout. However, there is never any comment or data showing a numerical or statistical analysis of this data. I would recommend that some quantification of fluorescence microscopy results be included – perhaps graphically or even just a statement of numbers of cells analyzed and percentage of a particular distribution observed.
Minor suggestions:
1) Lines 68 and 125: refs 22 and 23 I believe have little data on subcellular distribution – the authors’ have a 2013 FASEB paper that provides better support for the statements made.
2) Lines 130-131: “contaminants, reversed proteins, and proteins only identified by site”. It is not clear to me what these are or how they are defined.
3) Lines 142: The identification of TTLL5 is interesting. Is this the only TTLL expressed in these cells? Some comment on known expression in HEK cells could shed light on the implications of this finding.
4) Figure 3
a. The color scheme here seems unclear: both blue and green appear to be some shade of blue-green. While I would guess that the middle in fig 3a is the blue, it could be much clearer.
b. Figure 3b contains a lot of information. It is parenthetically referred to, but it seems not to be specifically discussed in the text. I am left wondering if it is significant, or if it provides any useful information above what is in 3a.
5) Line 208: PIBF is sometimes referred to as PIBF1.
6) Line 286: “a diffuse distribution with a punctate pattern” seems to be a contradiction. Diffuse or punctate?
7) Line 581. The sentence structure implied that the confocal microscope is a Duolink, from Sigma-Aldrich. Details of the microscope used should be provided here, not the supplier of the kit, which might go in line 566.
8) Section 4.5.2 does not describe “Image acquisition”, but rather describes the immunocytochemistry method.
Reviewer 3 Report
Reviewer’s Comments:
The manuscript “Proximity profiling of CCP6 identifies its association with centrosome organization and cilium assembly” is a very interesting work. In this work, the cytosolic carboxypeptidase 6 (CCP6) catalyzes the deglutamylation of polyglutamate side chains, a post-translational modification that affects proteins such as tubulins or nucleosome assembly proteins. CCP6 is involved in varied cell processes, such as spermatogenesis, antiviral activity, embryonic development, and pathologies like renal adenocarcinoma. In the present work, the cellular role of CCP6 has been assessed by BioID, a proximity labeling approach for mapping physiologically relevant protein-protein interactions (PPIs) and bait proximal proteins by mass spectrometry. We used HEK 293 cells stably expressing CCP6-BirA* to identify 41 putative interactors of this enzyme. This list of CCP6 proximal proteins displayed enrichment of proteins associated with the centrosome and centriolar satellites, indicating that CCP6 could be present in the pericentriolar material. In addition, we identified cilium assembly-related proteins as putative interactors of CCP6. The results are consistent with the data and figures presented in the manuscript. While I believe this topic is of great interest to our readers, I think it needs major revision before it is ready for publication. So, I recommend this manuscript for publication with major revisions.
1. In this manuscript, the authors did not explain the importance of the cilium assembly in the introduction part. The authors should explain the importance of cilium assembly.
2) Title: The title of the manuscript is not impressive. It should be modified or rewritten it.
3) Correct the following statement “By employing the proximity ligation assay (PLA), we showed that PCM1, PIBF1, and NudC are true CCP6 physical interactors. All in all, the proximal interactome of CCP6 confirms the association of CCP6 with centrosomes and centrioles, and a role in primary cilia formation and maintenance”.
4) Keywords: The cilium assembly is missing in the keywords. So, modify the keywords.
5) Introduction part is not impressive. The references cited are very old. So, Improve it with some latest literature like 10.3390/molecules27217368, 10.3390/molecules27207129
6) The authors should explain the following statement with recent references, “Among the identified partners, at least 14 proteins were associated with the centrosome, centriole, cilia biogenesis, and Joubert syndrome ciliopathy”.
7) Add space between magnitude and unit. For example, in synthesis “21.96g” should be 21.96 g. Make the corrections throughout the manuscript regarding values and units.
8) The author should provide reason about this statement “In brief, all replicate samples were grouped and LFQ intensities were log2 transformed. Proteins with less than 2 valid values in at least one group were removed and missing values were imputed from a normal distribution around the detection limit (with 0.3 spread and 1.8 down-shift)”.
9. Comparison of the present results with other similar findings in the literature should be discussed in more detail. This is necessary in order to place this work together with other work in the field and to give more credibility to the present results.
10) Conclusion part is very long. Make it brief and improve by adding the results of your studies.
11) There are many grammatic mistakes. Improve the English grammar of the manuscript.
